# Genome-Wide Association Study for Spot Blotch Resistance in Synthetic Hexaploid Wheat

**DOI:** 10.3390/genes13081387

**Published:** 2022-08-04

**Authors:** Nerida Lozano-Ramirez, Susanne Dreisigacker, Carolina P. Sansaloni, Xinyao He, José Sergio Sandoval-Islas, Paulino Pérez-Rodríguez, Aquiles Carballo Carballo, Cristian Nava Diaz, Masahiro Kishii, Pawan K. Singh

**Affiliations:** 1International Maize and Wheat Improvement Center (CIMMYT), Texcoco 56237, Mexico; 2Colegio de Post-Graduados, Montecillo 56230, Mexico

**Keywords:** foliar disease, spot blotch, genome-wide association study, synthetic hexaploid wheat, partial least squares regression

## Abstract

Spot blotch (SB) caused by *Bipolaris sorokiniana* (Sacc.) Shoem is a destructive fungal disease affecting wheat and many other crops. Synthetic hexaploid wheat (SHW) offers opportunities to explore new resistance genes for SB for introgression into elite bread wheat. The objectives of our study were to evaluate a collection of 441 SHWs for resistance to SB and to identify potential new genomic regions associated with the disease. The panel exhibited high SB resistance, with 250 accessions showing resistance and 161 showing moderate resistance reactions. A genome-wide association study (GWAS) revealed a total of 41 significant marker–trait associations for resistance to SB, being located on chromosomes 1B, 1D, 2A, 2B, 2D, 3A, 3B, 3D, 4A, 4D, 5A, 5D, 6D, 7A, and 7D; yet none of them exhibited a major phenotypic effect. In addition, a partial least squares regression was conducted to validate the marker–trait associations, and 15 markers were found to be most important for SB resistance in the panel. To our knowledge, this is the first GWAS to investigate SB resistance in SHW that identified markers and resistant SHW lines to be utilized in wheat breeding.

## 1. Introduction

Wheat (*Triticum aestivum* L.) is the most widely consumed food grain in the world. Global wheat production must therefore increase to meet the growing demand estimated for the next three decades [1]. It will be paramount to combine climate resilience, yield potential, and disease resistance in single wheat genotypes which could be grown across diverse environments. Known challenges that limit increased production rates are rapid climate change and emergence of new pathogenic variants. Foliar diseases, in particular, have become increasingly relevant for wheat in recent years, leading to significant losses in grain yield and quality [2]. Some of the factors driving foliar diseases are the commercial cultivation of susceptible varieties, the rapid evolution of causal pathogens, climate change, and unfavorable agricultural practices, which often lead to severe disease epidemics. About 21.5% of the global wheat production is lost each year to diseases [2], the majority of the losses attributed to fungal pathogens infecting multiple wheat organs such as root, stem, leaf, spike, and grain.

Spot blotch (SB) is caused by the fungus *Bipolaris sorokiniana* (Sacc.) Shoem syn. *Drechslera sorokiniana* (Sacc.) Subrm and Jain (syn. *Helminthosporium sativum*, teleomorph *Cochliobolus sativus*) and is considered one of the most destructive fungal diseases in humid and high temperature regions; they not only affect wheat, but also several other small grains worldwide such as barley, rye, and triticale [3,4,5,6,7,8,9]. The SB pathogen can infect all plant organs, but particularly leaves and grain; thus, reducing plant photosynthetic efficiency and grain quality. SB has a wide range of hosts among wild and cultivated Poaceae species [10,11,12]. SB symptoms are characterized by light to dark brown lesions on leaves, oval to elongated in shape [13], that extend and merge very quickly, resulting in tissue death.

The importance of SB in production losses has been widely documented. On average, yield loss of 15–20% due to SB has been reported in several countries under favorable climate conditions, yet the yield losses can reach up to 70% in susceptible varieties [14,15,16]. The growing threat of SB due to rising global temperatures and the accelerated evolution of pathogenic races have recently caught the attention of plant breeders and pathologists and created a sense of urgency for the identification of new sources of SB resistance.

The commercial cultivation of SB-resistant varieties is the most sustainable and cost-effective strategy to manage the losses incurred by SB [17,18,19]. Cultivar development for resistance to SB is slow due to the quantitative nature of resistance and a limited number of genes are known to have a major effect. Four SB resistance genes with major effects have been named to date, i.e., *Sb1* through *Sb4* [20,21,22,23]. Furthermore, several QTLs with minor effects have been found on almost all wheat chromosomes [24,25,26,27]. Most gene discovery studies undertaken to date have used biparental mapping populations, while a genome-wide association study (GWAS) using historical recombination usually provides a better resolution than bi-parental mapping. GWAS for resistance to SB found minor QTLs on chromosomes 2D, 3A, 4A, 4B, 5A, and 7B [28]; 1A, 1B, 1D, 4A, 5A, 5B, 6A, 6B, 6D, 7A, 7B [29]; and 1B, 3B 7B and 7D [30]. Recently, Bainsla et al. [31] found 25 marker–trait associations (MTAs) on 13 chromosomes explaining between 2.0 and 17.7% of the phenotypic variance. Tomar et al. [32] reported four new QTLs for resistance to SB in spring wheat on chromosomes 1A, 1D, 2B, and 6D. Most of the studies for resistance to SB concentrated on spring wheat, and only a few focused on winter wheat germplasm.

To identify novel and more effective sources, synthetic hexaploid wheat (SHW) (2*n* = 6x = 42; AABBDD), derived from a cross between *Triticum turgidum* L. (2*n* = 4x = 28; AABB) and *Aegilops tauschii* syn. *Ae. squarossa* (2*n* = 2x = 14; DD), could be an alternative source of resistance to SB as envisaged from other studies [33,34]. Previously, considerable levels of genetic variation were already recorded among SHW developed by the Wide Crosses Program of the International Maize and Wheat Improvement Center (CIMMYT) for different agronomic traits, disease resistance, and quality [33,35,36,37]. SHW was found to be promising in terms of resistance to SB and a few SHW lines showed better resistance than the resistant check variety ‘Mayoor’ [38].

Spot blotch is a major limiting factor for bread wheat production in hot and humid regions, particularly the Indo-Gangetic plains of South Asia. Despite the extensive breeding efforts, effective resistance to SB has not been observed in released cultivars, and the most promising cultivars have been found to be only partially resistant. Numerous studies have indicated that resistance to SB is polygenic, and multiple QTLs have been reported [24,26]. In CIMMYT, four biparental bread wheat populations were recently tested for SB resistance under Mexican environments, where several QTLs with minor effects were identified [24,25]. The same populations were further evaluated in South Asia with similar results, all QTLs presenting minor effects [26,27].

However, to our knowledge, no large-scale systematic screening and genetic study for SB resistance have been performed yet on SHW. Therefore, the objectives of this study were to (1) evaluate a set of 441 primary SHW lines for SB resistance under controlled environmental conditions and (2) to apply GWAS to identify potential new genomic regions of resistance that are not yet present in elite bread wheat germplasm.

## 2. Materials and Methods

### 2.1. Plant Material

A total of 441 SHW lines, generated by the CIMMYT’s Wide Crosses Program via hybridizing 40 durum wheat (DW) parents and 277 *Ae. tauschii* accessions, were used in this study. The DW parents were involved in 1–54 crosses and the *Ae. tauschii* accessions were used in 1–7 crosses (Appendix A). The SHWs were selected from a larger collection of 1524 SHWs for their resistance to diseases such as Fusarium head blight, Septoria tritici blotch, rusts, and have acceptable agronomic traits such as plant height and days to heading [34].

### 2.2. Phenotypic Evaluations of Spot Blotch

The disease screening was carried out in a greenhouse at CIMMYT, El Batán, Mexico (19°31’ N, 98°50’ W, elevation 2249 m above sea level) during 2018 and 2019. All 441 SHWs, along with the 40 DW parents and four checks including Chirya 3 (resistant), Sonalika and Ciano T79 (susceptible) and Francolin (moderately susceptible) were evaluated for SB resistance at the seedling stage, while the *Ae. tauschii* accessions could not be screened due to their nature and growth as a wild species. The seeds of SHW lines were vernalized to break down seed dormancy and to obtain an even germination. Experiments were planned in a randomized complete block design with six replicates for the SHW and eight replicates for the DW parents, with four plants per entry—grown in plastic containers as experimental units to obtain average values for their subsequent analysis. The size of the containers was 26.5 cm long, 20.5 cm wide, and 5 cm high. The seedlings were grown under controlled conditions with an ambient temperature of 22–25/16–18 °C (day/night) and with a 16 h photoperiod.

For disease expression, the isolate CIMFU 483 of Mexican *Bipolaris sorokiniana* (BSG40M2), a monosporic strain isolated from wheat collected in Agua Fria, Mexico, was used. This isolate is a ToxA producer, which was confirmed based on inoculation experiments with differential genotypes, infiltration experiments, and PCR with the ToxA1/ToxA2 primers. The isolate was grown in a 30% V8 media [39], and the conidia concentration for inoculation was adjusted to 7500 spores mL^−1^ using a Neubauer counting chamber. One drop of Tween 20 (a surfactant reagent) was added for every 100 mL of spore suspension.

Seedlings were inoculated at the two-leaf stage, when the second leaf was fully expanded, or two weeks after sowing. The seedlings were inoculated with a conidial suspension of the CIMFU 483 isolate until the leaves were at dew point. This inoculum was sprayed four times every 20–30 min using a hand sprayer. After the leaves dried, the trays were moved to a mist chamber (RH 100%, 22–24 °C) to promote infection. After 48 h, the plants were transferred back to the greenhouse bench. Seedling response was evaluated seven days post inoculation following the 1–5 ordinal lesion rating scale developed by Lamari and Bernier [40], which is based on the lesion type shown on the second leaf. The genotypes were grouped based on the mean score of replicates following 1.0–1.5 = Resistant (R); 1.6–2.5 = Moderately Resistant (MR); 2.6–3.5 = Moderately Susceptible (MS); and 3.6–5.0 = Susceptible (S).

### 2.3. Genotyping

Genomic DNA was extracted from the second leaf (0.25 mg per entry) of 10-day-old seedlings of each line of the SHW using the modified cetyltrimethyl ammonium bromide (CTAB) method described in the CIMMYT laboratory protocols [41]. The high-throughput genotyping method DArTseq^TM^ [42] was applied to all samples in the Genetic Analysis Service for Agriculture (SAGA) in CIMMYT, El Batan, Mexico.

Briefly, DArTseq is a complexity reduction method that includes two enzymes (PstI and HpaII) to create a genome representation of the set of samples. The PstI-RE site-specific adapter is tagged with 96 different barcodes, enabling the multiplexing of a 96-well microtiter plate with equimolar amounts of amplification products to run in an Illumina sequencer Novaseq6000 (Illumina Inc., San Diego, CA, USA). The successfully amplified fragments are sequenced with up to 83 bases, generating approximately 500,000 unique reads per sample. A proprietary analytical pipeline developed by DArT P/L was used to generate allele calls for SNP and presence/absence variation (PAV) markers [42]. A 100K consensus map [43] was used to obtain genetic positions of the SNPs in addition to the alignments to the reference genomes.

From the complete set of 441 SHW lines, 438 were genotyped and used for Genome Wide Association Study (GWAS). A total of 67,436 markers were scored, out of which 50% (34,790) could be aligned to reference genomes. Quality control was carried out based on the minimum lack of alleles, resulting in 5800 markers to be used for GWAS. The reference genomes used in this study were Chinese Spring IWGSC RefSeq v1.0 genome assembly [44] and durum wheat (cv. Svevo) Ref Seq Rel. 1.0 [45], along with the reference genome of *Ae. tauschii* (v.4, 2017) [46].

### 2.4. Statistical Analysis and Genome-Wide Association Study

For the disease data, statistical analyses were performed using the Statistical Analysis System version 9.1 [47]. An analysis of variance (ANOVA) was conducted on the average reactions of the SHW, the DW parents, and SB checks. The Best Linear Unbiased Estimates (BLUE) were computed for each of the 441 SHW genotypes and later used to conduct GWAS using the TASSEL (Trait Analysis by Association Evolution and Linkage) software ver. 5.2.73 [48].

The mixed linear model (MLM) by Yu et al. [49] was used to simultaneously include the level of relatedness based on marker data and identical by descent (IBD) computed from the coefficient of parentage, which controls population structure. Additionally, population structure was controlled by fitting the first five principal components (PC) from the kinship matrix taken as the fixed variate and the coefficient of parentage (COP) as the random variable. The false-discovery rate (FDR) was used to assess the significance of the *p*-value (<0.05) [49]. The allelic effects of the significant MTAs were estimated as the difference between the mean value of lines, with and without the favorable alleles, and were presented as box plots.

### 2.5. Partial Least Squares Regression

We used the Partial Least Squares (PLS) method to apply the results of GWAS analyses to practical application to breeding. Extensive studies to assess the importance of environmental and genotypic covariables in multi-environment plant breeding trials were carried out using the PLS method [50,51,52,53].

In the context of this study, the PLS relates in a single estimation procedure (1) the two-way table of phenotypic measurements of SB of the SHW lines in 6 replicates in the greenhouse (and on the mean across the six replicated) and (2) the total number of significant markers found in the current GWAS study (41 explanatory variables). PLS regression describes explanatory (markers) as linear combinations of the complete set of measures of SB on SHW cultivars with no limit to the number of marker covariables or to the number of SHW lines that can be used.

## 3. Results

### 3.1. Resistance to Spot Blotch at the Seedling Stage

The SB development observed during seedling evaluation in the greenhouse was even and consistent. ANOVA showed significant differences among SHWs (*p* < 0.001). The checks Chirya 3, Sonalika, Ciano T79, and Francolin displayed scores of 1.4, 4.0, 4.0, and 2.8, respectively (Table 1), verifying the identity of the *B. sorokiniana* isolate used and a successful inoculation.

Most of the 441 SHW lines displayed resistant and moderately resistant reactions (Appendix A), i.e., 250 (56.7%) showed resistance (R) and 161 (36.5%) showed moderate resistance (MR) reactions with disease scores of 1.0–2.5, comparable to the resistant check Chirya 3. Only 30 SHWs (6.8%) were moderately susceptible (MS) or susceptible (S) with disease scores of 3.0–4.1. These scores were still lower than the scores of the susceptible checks, Sonalika, and Ciano T79 (Table 1 and Figure 1).

The SB reaction of DW parents revealed that 18 (45%) parents had reaction scores of 1.0–1.5 (R) and 14 (35%) reaction scores of 1.6–2.5 (MR), developing mostly small dark to maroon lesions on those that had extended 1–2 mm in length with chlorotic edges during the initial infection. Eight entries (20%) were observed to have a mean reaction score between 2.6 and 3.6, being considered moderately susceptible (MS) to susceptible (S), whereas the leaves were observed to die/senescence when the light brown to dark brown oval to elongated blotches extended and merged very quickly (Table 1 and Appendix A). The SB reaction scores of the DW parents compared to the scores of the SHW indicated that the SB resistance of SHW was likely inherited from both DW and *Ae. tauschii* parents.

### 3.2. Genome-Wide Association Study Using Different References Genomes

The first two principal components (PCs) based on the DArTSeq markers separated two clear groups of entries of similar sizes and some entries in between, explaining around 34% of the total variability. This population structure was controlled by fitting the first five PCs derived from the correlation matrix as fixed covariates. Additionally, the coefficient of parentage used as a random variable to fit the GWAS mixed linear model (MLM) effectively controlled the remaining population structure after fitting the first five PCs.

From the complete set of 441 SHW lines, 438 were genotyped and used for the Genome-Wide Association Study (GWAS). A total of 67,436 markers were scored, out of which 50% (34,790) could be aligned to reference genomes. Quality control was carried out based on the minimum lack of alleles, resulting in 5800 markers to be used for GWAS.

Out of the DArTSeq markers that could be aligned to the whole genome sequence of cv. Chinese Spring (CS, IWGSC RefSeq v1.0), 20 significant MTAs were identified as shown in Appendix A and Figure 2, being located on chromosomes 1B (1), 1D (1), 2A (1), 2D (3), 3A (2), 3B (1), 3D (1), 4A (1), 5A (2), 5D (2) 6D (1), 7A (2), and 7D (2). The markers with the highest allele substitution effects were located on chromosomes 7D (1.11), 3A (0.33), and 5D (0.32).

Looking at the markers located on the 100 K consensus map, 32 significant MTAs were detected, as shown in Appendix A and Figure 3, and found to be located on chromosomes 1B (7), 1D (2), 2A (2), 2B (3), 2D (2), 3B (2), 3D (2), 4A (3), 4D (1), 5A (2), 5B (1), 6B (1) 7A (3), and 7B (1). The markers with the highest allele substitution effects were located on chromosomes 5B (1.12), 3B (0.53), and 2B (0.24). Nine MTAs based on the IWGSC Ref Seq v1.0 overlapped with those presented in Appendix A. Therefore, three MTAs showed the same chromosome allocation on the genetic and physical maps, while six MTAs showed different chromosome assignments (yet mainly homologous chromosomes) on both maps.

When markers aligned to the DW cultivar Svevo and the *Ae. tauschii* reference genomes were considered, 10 MTAs were identified on chromosomes 1B (1), 2A (1), 2B (1), 2D (1), 3A (2), 3B (2), 4D (1), and 7A (1) (Appendix A and Figure 4). However, only three markers in Appendix A coincided with those found in Appendix A. Marker ID 1240012 on chromosome 2B in Svevo was found to be on chromosome 7D when aligned to the physical map of CS and on chromosome 5B in the 100K consensus map. The markers with the highest allele substitution effects ranged from 1.10 (2B), 0.33 (3A), to 0.16 (3A).

Overall, a total of 41 genomic regions identified using the different maps are summarized in Table 2. A re-alignment of the marker sequences to the ABD, AB, and D genomes verified the physical position of several of the significant SNPs and could identify their physical positions across species. However, among all, 11 MTAs could not be assigned positions on the physical map. Furthermore, 23 MTAs were found within annotated high-confidence gene sequences, with 10 of these 23 candidate genes annotated in the CS reference genome, 6 in Svevo reference genome, and 7 in the *Ae. tauschii* reference genome (Appendix A). These significant MTAs were detected on 15 chromosomes with the maximum number of 5 MTAs on chromosome 1B and 1 each on 6D and 7B, and their R^2^ values varied from 0.03 to 0.07. Among the five markers detected on chromosome 1B, the highest R^2^ value of 0.06 was found for marker ID 1145134 that is in proximity with marker ID 5582520, with two other markers (IDs 4261287 and 7335825) distal to them and one (ID 100033209) proximal to them. Three MTAs were found on chromosome 2A, with marker ID 1144884 exhibiting the highest R^2^ value of 0.07. Two MTAs on chromosome 5A (IDs 3570010 and 1046932) were found with low R^2^ values of 0.03 for each one. Allelic effects ranged from 0.01 to 1.11 for the MTAs on 4D (ID 2243087) and 7D (ID 1240012), respectively.

### 3.3. Identified MTA

On chromosome 1B, the reported positions for five MTAs showed two MTAs (markers 4261287 and 7335825) nearby, at 51.3 and 52.6 cM, and two MTAs (markers 5582520 and 1145134) at 96.9–98.0 cM, respectively, resulting in three different QTLs identified for SB on chromosome 1B. On chromosome 2D, two MTAs (markers 1122278 and 2243785) were positioned 11.02 Mbp apart but with an R^2^ of 0.08 and a probability of linkage disequilibrium (LD) of 1.23 × 10^−7^ forming a third MTA. Additionally, two markers on chromosome 3A with a distance of only 0.11 Mbp (markers ID 1019955 and 474554774) showed a linkage disequilibrium R^2^ of 0.8138, with a *p*-value of 1.21 × 10^−7^. The two significant markers on 3D were located at a distance of 20 cM; thus, being considered unlinked. On chromosome 4A, markers 1162615 and 100036641 were mapped near each other, at 96.1 and 96.4 cM, respectively, and thus could be considered one single MTA.

### 3.4. Frequency of Resistance Alleles within Individual SHWs

The frequency of resistance alleles in the SHWs was examined with the aim of identifying lines with high numbers of resistance alleles to be used for further resistance breeding. A total of 59 SHW lines carried more than 30 of the 41 identified resistance alleles with an average SB score of 1.3 (Figure 5). Although not shown in this figure, there are 32 SHW lines with >32 resistance alleles and 15 SHW lines with >34 resistance alleles, which could be the top candidates for further evaluation and breeding. SHW lines with less resistance alleles (<16 R alleles) showed increased susceptibility and demonstrated the additive nature of the resistance alleles.

### 3.5. Interpretation of Results from Partial Least Squares

The results of the PLS are shown in Figure 6, where the first two PLS factors explained around 26% of the total variability, and 15 molecular markers (green color) with a frequency of R alleles greater than 84% and 32 SHW lines (red color) having more than 32 resistance alleles (Figure 6). The arrows from the center to the upper-left quadrant show the six phenotype measurements of SB (SB1-6) and their overall mean (Mean SB). The SHW lines are distributed in a linear manner from the lower-right quadrant (more resistance lines) to the upper-left quadrant (more susceptible lines). The 15 markers were located at the center and on the right-hand side of the biplot (green letter-numeric combination), and the 32 most resistant SHW lines (red numbers) are located towards the lower-right quadrant. From a practical breeding perspective, the 15 markers and the 32 SB resistance lines could be prioritized in crosses between SHW lines and elite bread wheat lines in breeding and pre-breeding programs.

## 4. Discussion

Genome-wide association studies were performed to uncover SNP markers related to SB resistance in bread wheat. One such study was conducted by [54] on 528 spring wheat accessions for seedling resistance against SB, and 11 MTAs were identified. The same panel was analyzed earlier by [30], but only four genomic regions were identified, due to fewer markers being used, emphasizing the importance of high-density marker data. A recent GWAS was reported by [55], who studied a total of 6736 CIMMYT breeding lines for SB resistance in field experiments conducted throughout several years (2014–2019), and up to 214 MTAs were identified in at least one year, 96 were repeatable in at least two years and all had minor effects.

To our knowledge, to date no GWAS has been reported on SB resistance in SHW, although several studies reported good resistance of SHW to SB. In earlier studies, *Ae. tauschii* was used to transfer potential SB-resistant genes through *T. turgidum* × *Ae. tauschii* or *T. aestivum* × *Ae. tauschii* crosses [35]. Diverse *Ae. tauschii* accessions were used to make SHW lines, which exhibited promising SB resistance and often performed better than the resistant check Mayoor [38]. A series of SHW was developed and then screened for several biotic and abiotic stresses, and promising entries were either used for commercial cultivars or as pre-breeding materials to develop new genotypes. The authors of [33] reported eight SHW accessions with SB resistance, along with sources of resistance to other diseases.

Our study revealed that the evaluated SHWs displayed a considerable resistance to SB, with 38% of the SHW lines showing better resistance than the resistant check Chirya 3. According to the pedigree information, SB resistance of the panel might be based on diverse DW and *Ae. tauschii* backgrounds and was thus likely contributed by multiple SB resistance genes that was in agreement with the GWAS results.

### 4.1. Novelties of the Significant Markers Found in the Current Study

Previous genetic studies have identified a range of SB resistance genes/QTL, residing on all wheat chromosomes except 4D and 5D, as summarized recently by [56]. Some of these loci exhibited major effects, such as the nominated *Sb* genes, yet most of them showed minor effects. The same applies to the current study, where a total of 41 significant markers on 15 chromosomes were found to be associated with SB resistance, and none of them showed any major effects. This again confirmed the polygenic nature of SB resistance described in previous studies [24,26,55]. The significant MTAs were identified on AB genome chromosomes as well as on D genome chromosomes, suggesting that SB resistance in the SHWs was derived from both their DW and *Ae. tauschii* parents.

MTAs were identified on all seven D genome chromosomes, especially chromosomes 4D and 5D, on which no QTL/MTA has been reported so far [56]; thus, confirming their novelty. The two MTAs on chromosome 4D were located on short arm (marker 2243087) and long arm (marker 3023637); on chromosome 5D the physically distant markers must represent two different QTL. MTAs on chromosomes 1B (marker 1145134), 2D (marker 1122278 and 2243785), 3A (marker 1019955 and 2279238), and 6D (marker 1698662) also suggested to be novel since no QTL/MTA has been reported in the vicinity of these markers [56].

However, some MTAs were found within known QTL regions. For example, the two MTAs on chromosome 1BS (markers 4261287 and 7335825) were in close proximity to the MTAs reported by [29]. Likewise, on chromosome 3B, marker 4992362 was closely located to an MTA reported by [31]. Nevertheless, close linkage or coincidence does not necessarily mean that the identified regions represent the same QTL/MTA, especially because our study screened SHW, while those published previously evaluated common wheat. It is noteworthy that some markers did not show any BLAST hit on the three reference genomes, e.g., marker 7335825 on chromosome 1B and marker 7492146 on chromosome 2B. These MTAs represent variants absent in the reference genomes and might be worthy of further investigation.

### 4.2. Candidate Genes for the Identified Marker–Trait Associations

The significant markers identified from the GWAS were further evaluated for their association with disease resistance-related genes. We identified 23 plant defense-related protein families across multiple chromosome regions, of which only 13 have a known protein function. For example, marker 12779374 on chromosome 1D was identified within the gene TRITD1Bv1G224330 (Appendix A and Table 2), which is involved in the synthesis of the lectin receptor kinase that has an important function for the general immunity of the plants [57]. Similarly, marker 1240012 on 7D was located within the gene TRITD2Bv1G075350 related to protein U-box domain containing protein 4, associated with the control of grain production [58]. However, it should be noticed that these candidate genes might not be the underlying genes for the MTAs, due to the usually large linkage disequilibrium blocks in the wheat genome [59].

Furthermore, marker 1283998 on chromosome 3B marked an SNP within gene TRITD3Bv1G194800, which is a protein described as disease resistance protein RPM1 G, again involved in the general resistance of plants to various diseases [60]. Marker 4992362 on chromosome 3B marked the gene TRITD3Bv1G257410, which is identified as protein Serpin that participates in the regulation of proteolytic complex systems [61], whereas marker 1011260 (in chromosome 3D) falls within the gene TraesCS3D02G407000, a peroxidase protein that has the divergence role in different pathogens systems in plants [62]. Furthermore, marker 100016153, aligned on chromosomes 5A and 5D, was located within the genes TraesCS5A02G146400 and TRITD5Av1G111170, in which two proteins, Mannan endo-1,4 -beta-mannosidase 6 and Mannan endo-1,4-beta-mannosidase-like protein, are involved.

Note that marker 4002611 on chromosome 7A did fall within the gene TRITD7Av1G003410, a Pectin lyase-like superfamily protein, which has an important role in the development and maturity process of the plant. This protein also acts on the peptic substances presented as structural polysaccharides in the primary cell walls of the superior plants [63]. Marker 22765212, on chromosome 7D, was included in gene TraesCS7D02G278500, which is found in the ribosomal protein that plays a fundamental integral role in the growth and development of the plant, as well as participating in the general defense mechanism of the plants [64].

### 4.3. Application of GWAS for Use in Practical Breeding

Genome-wide association studies (GWAS) are a powerful option for the genetic characterization of quantitative traits and have been widely used to analyze agronomic and disease traits. With the increasing number of diseases affecting cultivated wheat plants, the option of developing resistance SHW lines has been widely used. This is the first GWAS study to assess significant MTA of SB from a diverse collection of 441 SHW lines, and 41 significant markers and a range of SHW lines with high SB resistance were identified. In the PLS analysis, a subset of markers and SHW lines were identified that are more suitable for future breeding and pre-breeding activities.

Results of this study showed 15 molecular markers with a frequency of R alleles greater than 84% and 32 SHW lines having more than 32 resistance alleles. The PLS plot show the specific locations of the 15 markers and the 32 most resistant SHW lines. From a practical breeding perspective, these markers with R alleles and the SB resistance lines could be used in future breeding crosses.

## 5. Conclusions

This is the first GWAS study to investigate MTAs for SB resistance in a diverse collection of 441 SHW lines from CIMMYT. GWAS found a total of 41 significant markers related to SB resistance, being distributed on 15 wheat chromosomes, and many of them are novel. We were able to identify highly resistant SHW lines with most resistance alleles of the significant markers that can be used in wheat breeding programs.

## Figures and Tables

**Figure 1 genes-13-01387-f001:**
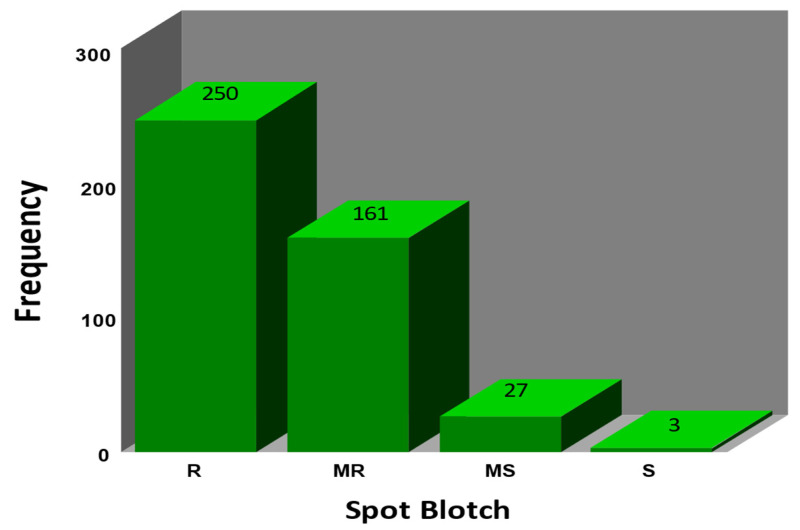
Histogram of spot blotch (SB) scores for different reaction types, which include Resistant (R, 1.0–1.5), Moderately Resistant (MR, 1.6–2.5), Moderately Susceptible (MS, 2.6–3.5), and Susceptible (S, 3.6–5.0).(data extracted from Appendix A).

**Figure 2 genes-13-01387-f002:**
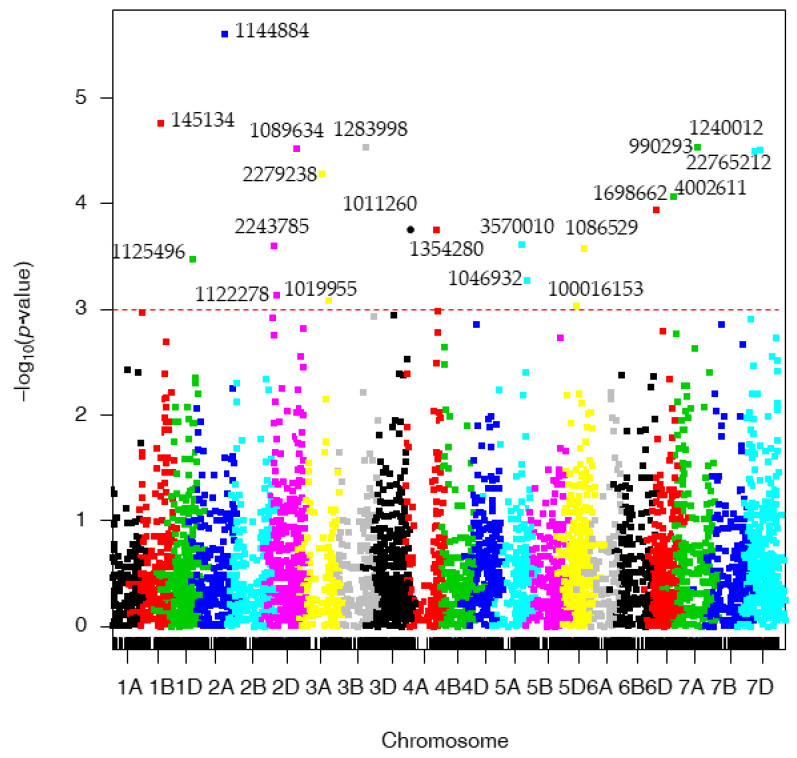
Manhattan plots for spot blotch (SB) disease corresponding to the physical position of Chinese spring Ref Seq ver.1.0. The *p*-values are shown on a log_10_ scale. The marker is considered significant if log_10_ scale is 3 or higher.

**Figure 3 genes-13-01387-f003:**
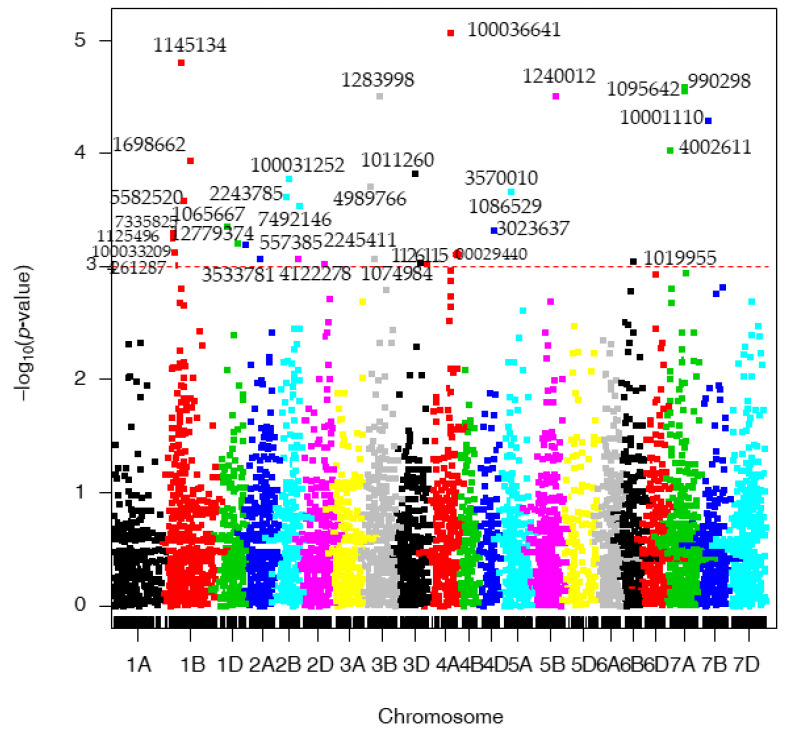
Manhattan plots for spot blotch disease (SB) corresponding to the consensus map. The *p*-values are shown on a log_10_ scale. The marker is considered significant if log_10_ scale is 3 or higher.

**Figure 4 genes-13-01387-f004:**
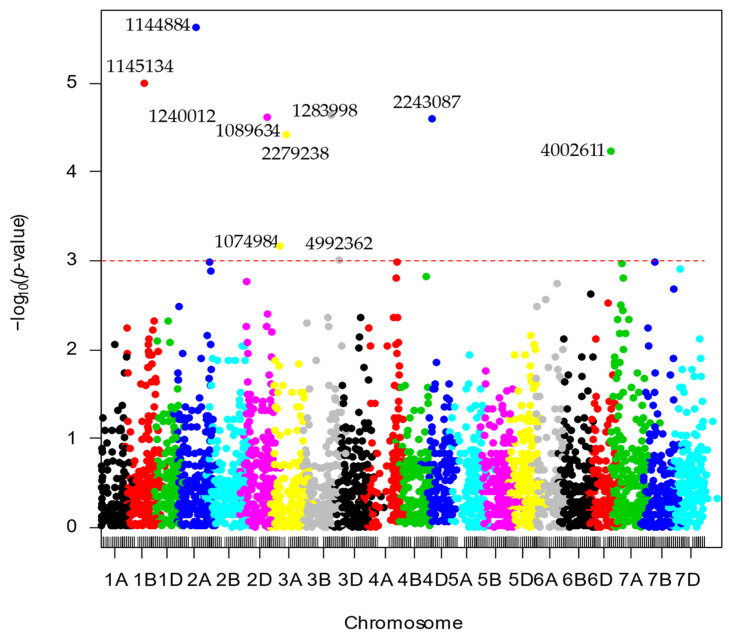
Manhattan plots for spot blotch (SB) disease corresponding to the durum wheat (cv. Svevo) and *Ae. tauschii* reference genomes (Ref Seq Rel. 1.0). The *p* values are shown on a log_10_ scale.

**Figure 5 genes-13-01387-f005:**
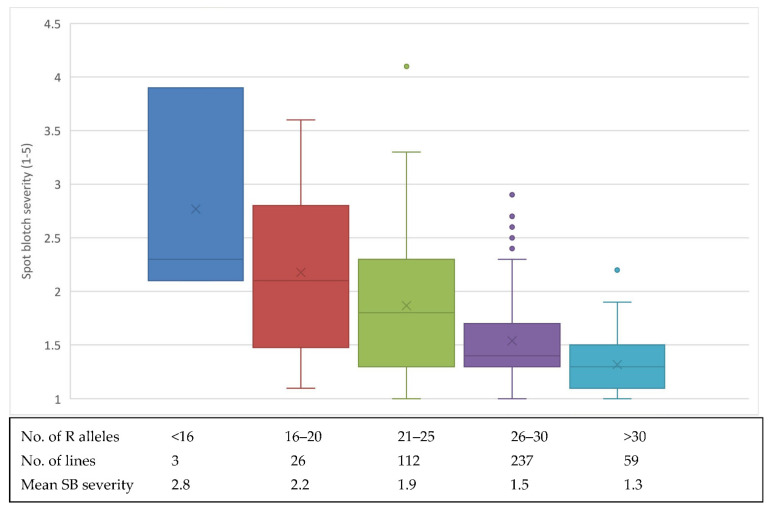
Boxplot showing the effects of stacking different number of resistance (R) alleles (QTL) on mean SB severity. The average severity is represented by the ‘x’ symbol and the median by the horizontal line inside.

**Figure 6 genes-13-01387-f006:**
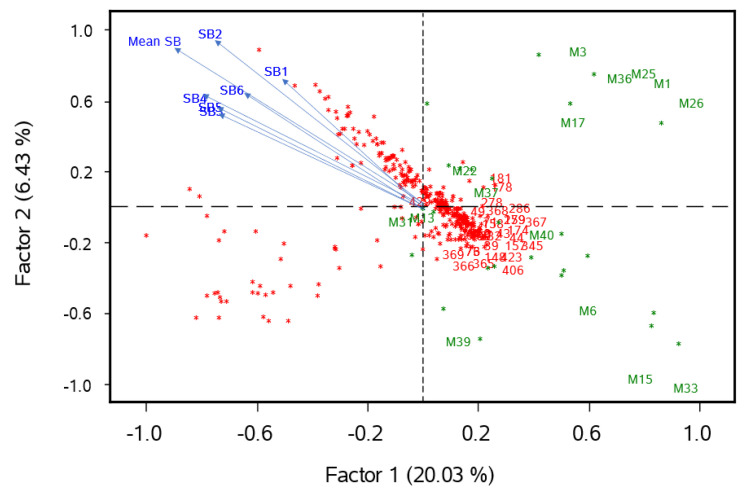
Biplot chart showing the first two PLS factors for 41 significant markers and 438 SHW lines, where SB measured in the greenhouse in six replicates (SB1-6) and overall mean (Mean SB) are shown (lines from the center to the upper right quadrant). The 15 molecular markers with a frequency of resistance alleles greater than 84% were M1 (4261287 chr1B), M3 (5582520 chr1B), M6 (1065667 chr1D), M13 (100031252 chr2B), M15 (2243785 chr2D or chr2B), M17 (1019955 chr3A or chr6B), M22 (1074984 chr3D), M25 (1162615 chr4A), M26 (100036641 chr 4A), M31 (1046932 chr 5A), M33 (1086529 chr5D or chr5A), M36 (1095642 chr7A), M37 (990293 chr7A), M39 (2245411 chr7D or chr2D), and M40 (1240012 chr7D or chr5B) (marker IDs are presented in Table 2). The 32 SHW lines having more than 32 resistance alleles are identified with red numbers. The remaining markers and SHW lines are represented by green and red dots, respectively.

**Table 1 genes-13-01387-t001:** Spot blotch (SB) reactions of 40 durum wheat (DW) parents, their respective synthetic hexaploid wheat (SHW) and four checks. Reactions are defined as Resistant (R, 1.0–1.5), Moderately Resistant (MR, 1.6–2.5), Moderately Susceptible (MS, 2.6–3.5), and Susceptible (S, 3.6–5.0). For 18 SHW lines, their DW parents were not identified.

Pedigree of the DW Parents	DW Parents		Number of Progeny (*Ae. tauschii*)	SHW	
	SB Score	Reaction Type		Mean SB Scores	Mean Reaction Type
68.111/RGB-U//WARD	3.6	S	7	1.7	MR
68.111/RGB-U//WARD RESEL/3/STIL	3.4	MS	1	1.5	R
68.111/RGB-U//WARD/3/FGO/4/RABI	3.2	MS	31	1.5	R
68112/WARD	3.2	MS	3	1.3	R
6973/WARD.7463//74110	3.1	MS	13	1.3	R
ACONCHI 89	3.0	MS	2	1.7	MR
ALG86/4/FGO/PALES//MEXI_1/3/RUFF/FGO/5/ENTE	2.7	MS	31	1.5	R
ALTAR 84	2.6	MS	1	2.3	MR
ARLIN_1	2.5	MR	4	1.6	MR
BOTNO	2.5	MR	30	1.8	MR
CERCETA	2.4	MR	13	1.5	R
CHEN_7	2.3	MR	4	1.5	R
CPI8/GEDIZ/3/GOO//ALB/CRA	2.3	MR	12	1.4	R
CROC_1	2.3	MR	3	1.7	MR
D67.2/PARANA 66.270	2.3	MR	31	1.8	R
DECOY 1	2.2	MR	1	1.8	MR
DVERD_2	2.1	MR	4	1.6	MR
FALCIN_1	2.0	MR	39	1.5	R
FGO/USA2111	2.0	MR	3	1.9	MR
GAN	1.9	MR	7	2.1	MR
GARZA/BOY	1.8	MR	3	2.6	MS
GREEN	1.7	MR	4	2.0	MR
KAPUDE_1	1.5	R	13	1.6	MR
LARU	1.4	R	30	1.5	R
LCK59.61	1.3	R	54	1.8	MR
LOCAL RED	1.3	R	4	1.5	R
RABI//GS/CRA	1.3	R	6	1.9	MR
RASCON	1.3	R	1	1.0	R
ROK/KML	1.3	R	20	1.8	MR
SCAUP	1.3	R	7	1.9	MR
SCOOP_1	1.2	R	3	1.2	R
SCOT/MEXI_1	1.2	R	2	1.2	R
SHAG_22	1.1	R	2	1.9	MR
SNIPE/YAV79//DACK/TEAL	1.1	R	4	1.3	R
SORA	1.1	R	1	1.4	R
STY,DR/CELTA//PALS/3/SRN_5	1.1	R	14	1.6	MR
TK SN1081	1.0	R	7	1.3	R
YAR	1.0	R	5	1.8	MR
YARMUK	1.0	R	4	1.9	MR
YAV_2/TEZ	1.0	R	1	1.3	R
Chirya 3 (R check)	1.3	R	-	1.4	R
Sonalika (S check)	4.2	S	-	4.0	S
Ciano T79 (S check)	4.3	S	-	4.0	S
Francolin (MS check)	2.7	MS	-	2.8	MS

**Table 2 genes-13-01387-t002:** Significant marker–trait associations for seedling resistance to spot blotch, their position in different reference genomes, associated candidate genes, and GWAS statistics. The table contains the physical position based on Chinese Spring (CS) reference genome, the chromosome and the genetic position based on cM, the BLAST results against the CS, Svevo, and *Ae. tauschii* reference genomes, genes, freq. of resistance markers, *p*-values, Marker R^2^. −log_10_ *p*-values and effect of allele.

Chr.	Marker ID	Physical Position (CS) Ref Seq v1.0)	Chr	Genetic Position (cM)	BLASTN to I WGSC Ref Seq V1.0	BLAST to Ref Seq Svevo	BLAST to Ref Seq *Ae. tauschii*	Gene (s)	Frequency of Resistance Marker Allele	*p*-Value	Marker R^2^	−log_10_ *p*-Value	Effect of Allele
1B	4261287		1B	51.29	1B: 17,537,160–17,537,233	no good hit found	no good hit found		0.88	9.83 × 10^−4^	0.04	3.01	−0.29
1B	7335825		1B	52.56	no good hit found	no good hit found	no good hit found		0.83	4.96 × 10^−4^	0.04	3.30	−0.19
1B	5582520		1B	96.91	no good hit found	no good hit found	no good hit found		0.89	2.70 × 10^−4^	0.04	3.57	−0.26
1B	1145134	406039536	1B	98.03	1B: 406,039,533–406,039,608	1B: 399,260,866–399,260,941			0.63	1.64 × 10^−5^	0.06	4.79	−0.05
1B	100033209		1B	139.32	no good hit found	no good hit found	no good hit found		0.83	8.35 × 10^−4^	0.04	3.08	−0.66
1D	1065667		1D	12.27	1D: 6,248,618–6,248,679		1D: 6,917,141–6,917,202		0.94	4.50 × 10^−4^	0.04	3.35	0.23
1D	1125496	416590812	1B	51.289	1D: 416,590,808–416,590,883		1D: 424,102,922–424,102,997	AET1Gv20777500	0.82	3.36 × 10^−4^	0.03	3.47	NaN
1D	12779374		1D	130.64	1D: 486,387,813–486,387,877	1B: 667,753,290–667,753,354	1D: 493,826,928–493,826,992	TraesCS1D02G441400 AET1Gv21021400 TRITD1Bv1G224330	0.12	6.25 × 10^−4^	0.04	3.20	0.00
2A	5573285		2A	45.45	no good hit found	no good hit found	no good hit found		0.78	5.74 × 10^−4^	0.04	3.24	0.17
2A	1144884	583026867			2A: 583,026,863–583,026,938	2A: 576,091,990–576,092,065			0.77	2.50 × 10^−4^	0.07	5.60	0.02
2A	3533784		2A	123.66	aligns only to 2B	2A: 77,422,937–77,422,941			0.64	9.75 × 10^−4^	0.04	3.01	−0.13
2B	7492146			107.03	no good hit found	no good hit found	no good hit found		0.83	3.01 × 10^−4^	0.04	3.52	0.24
2B	100031252			55.48	no good hit found	no good hit found	no good hit found		0.88	1.66 × 10^−4^	0.04	3.78	NaN
2D	1122278	21621448	2D	20.85	2D: 21,621,445–21,621,520		2D: 22,832,366–22,832,441	TraesCS2D02G054200	0.61	8.39 × 10^−4^	0.04	3.08	−0.14
2D	2243785	32640660	2B	40.74	2D: 32,640,657–32,640,732		2D: 33,858,967–33,859,042	TraesCS2D02G076500	0.86	2.46 × 10^−4^	0.04	3.61	−0.18
2D	1089634	509231294			2D: 509,231,291–509,231,366		2D: 507,788,059–507,788,134	AET2Gv20890600	0.05	3.10 × 10^−4^	0.05	4.51	0.03
3A	1019955	474447292	6B	46.69	3A: 474,447,288–474,447,363	3A: 477,078,635–477,078,710			0.92	9.28 × 10^−4^	0.04	3.03	−0.46
3A	2279238	474554774			3A: 474,554,770–474,554,845	3A: 477,190,300–477,190,375			0.84	5.27 × 10^−5^	0.05	4.28	0.33
3B	4989766		3B	19.56	no good hit found	no good hit found	no good hit found		0.81	1.97 × 10^−4^	0.04	3.71	0.53
3B	1283998	593544135.00	3B	68.53	3B: 593,544,132–593,544,207	3B: 593,903,780–593,903,855		TRITD3Bv1G194800	0.10	3.04 × 10^−5^	0.05	4.52	−0.02
3B	4992362	775474348.00			3B: 763,236,117–763,236,191	3B: 775,474,345–775,474,420		TraesCS3B02G520000 TRITD3Bv1G257410	0.21	9.91 × 10^−4^	0.04	3.00	0.02
3D	1074984		3D	61.81	3D: 401,883,953–401,884,028		3D: 409,258,183–409,258,258	TraesCS3D02G291900 AET3Gv20689000	0.86	9.10 × 10^−4^	0.04	3.04	0.17
3D	1011260	520678096	3D	82.16	3D: 520,678,093–520,678,168		3D: 529,110,490–529,110,565	TraesCS3D02G407000 AET3Gv20921800	0.21	1.83 × 10^−4^	0.04	3.74	−0.05
4A	1351280	629433955.00			4A: 629,433,952–629,434,027	4A:623,641,790–623,641,858		TraesCS4A02G355400	0.84	1.78 × 10^−4^	0.04	3.75	−0.06
4A	1162615		4A	96.08	4A: 661,535,726–661,535,794	4A:661,278,198–661,278,266			0.87	9.57 × 10^−4^	0.04	3.02	−0.26
4A	100036641		4A	96.36	no good hit found	no good hit found	no good hit found		0.92	8.42 × 10^−6^	0.06	5.07	−0.39
4A	100039440		4A	113.91	aligns to many chromosomes but less than 100%	4A:693,427,125–693,427,193 4A:693,425,785–693,425,853			0.83	8.99 × 10^−4^	0.04	3.05	−0.32
4D	3023637	474561316	4D	66.12	no good hit found	no good hit found	no good hit found		0.05	4.86 × 10^−4^	0.04	3.31	−0.02
4D	2243087	54178331			4D: 51,304,835–51,304,903		4D:54,178,332–54,178,400		0.07	2.61 × 10^−5^	0.05	4.58	0.01
5A	3570010	521764788	5A	36.99	5A: 521,764,784–521,764,859	5A:484,938,946–484,939,014			0.02	2.40 × 10^−4^	0.03	3.62	NaN
5A	1046932	622389460			5A: 622,389,461–622,389,529 4A:552297214–552297282	5A:583,637,584–583,637,652 4A:545,007,545–545,007,613			0.85	5.52 × 10^−4^	0.03	3.26	NaN
5D	100016153	232599413			5D: 232,599,413–232,599,475 5A:322,677,280–322,677,342	5A:316,073,030–316,073,092	5D:246,553,454–246,553,516	AET5Gv20379200, TraesCS5A02G146400 TRITD5Av1G111170	0.72	9.35 × 10^−4^	0.04	3.03	0.32
5D	1086529	410253879	5A	36.99	5D: 410253875–410253950		5D:418,190,498–418,190,566		0.89	2.65 × 10^−4^	0.04	3.58	0.21
6D	1698662	42940457.00	1B	148.15	6D: 42,940,453–42,940,522		6D: 64,808,834–64,808,903		0.76	1.13 × 10^−4^	0.05	3.95	−0.27
7A	4002611	7938756.00	7A	7.25	7A:7,938,757–7,938,825	7A:6,228,579–6,228,647		TraesCS7A02G019400 TRITD7Av1G003410	0.10	8.88 × 10^−5^	0.05	4.05	−0.04
7A	1095642		7A	75.85	no good hit found	no good hit found	no good hit found		0.88	2.90 × 10^−5^	0.05	4.54	−0.29
7A	990293	621213334.00	7A	88.42	4A:142,973,443–142,973,511 7A:621,213,334–621,213,402	4A:140,470,489–140,470,557 7A:616,593,441–616,593,509			0.85	3.11 × 10^−5^	0.05	4.51	−0.03
7B	100011110		7B	46.26	no good hit found	no good hit found	no good hit found		0.84	5.27 × 10^−5^	0.05	4.28	−0.23
7D	2245411		2D	118.19	7D: 69,417,014–69,417,082		7D: 70,389,436–70,389,511		0.89	9.54 × 10^−4^	0.04	3.02	−0.14
7D	1240012	150762254	5B	98.36	7D: 150,762,250–150,762,325	2B: 196,456,606–196,456,681	7D: 151,389,082–151,389,157	TRITD2Bv1G075350	0.89	3.19 × 10^−5^	0.05	4.50	1.11
7D	22765212	268565893			7D: 268,565,890–268,565,965		7D: 270,502,277–270,502,352	TraesCS7D02G278500 AET7Gv20675900	0.05	3.15 × 10^−5^	0.05	4.50	0.02

## Data Availability

The original contributions presented in the study are publicly available.

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
