# Peer review of "Genome-Wide Association Study for Spot Blotch Resistance in Synthetic Hexaploid Wheat"

_genes, 2022, doi:10.3390/genes13081387_

Round 1

Reviewer 1 Report

Overall, the research manuscript is well crafted, however, there are minor grammatical errors (text editing) which have been pointed out. The conclusion part has a scope of improvement (presenting the names of MTAs, chromosomes and  entries) which will help improve the reach and visibility of the published paper.

Author Response

Dear Sir/Madam

Thanks for your comments. We have revised the manuscript based on your comments and made significant improvement in English (edited by profession communication expert).

Please see the highlighted inputs in the revised manuscript for your review.

Thanks and regards

Pawan

Reviewer 2 Report

Comments to authors:

Overall, this is a good study for identifying spot blotch resistance in a panel of synthetic hexaploid wheat. You found significant markers and the lines contained multiple markers. However, the description of methods and the presentation of results require improvement. The discussion needs to be revised as well. Many sentences could be revised for reading smoothly. Please see my comments and suggestions. 

Line 14, if two SHW lines did not germinate (line 194), there were only 441 lines that were evaluated.

Line 53, you may delete “disease”.

Line 56, new sources of “what”?

Line 71-72, what is the relevance to have this sentence?

Line 83, still, please clear the number of lines.

Line 104, the size of containers?

Line 117-118, how many ml of suspension per seedling? How long was between each time spray?

Line 128, how many grams of leaf? Did you use the same plants for both phenotyping and genotyping?

Line 133-140, if this is the description of DArTSeq, it should be in introduction section. Please clarify the procedure you performed for genotyping.

Line 143-146, this should be in result section.

Line 157, it is not proper to cite reference like “by [49]”.

Line 157-172, again, too much description about MLM and PLS. Please describe in detail about how you did instead of introducing them.

Line 184, please replace “between” with “among”.

Figure 1, I don’t think that “Frequency” and “Spot Blotch” are suitable titles. The legend should have more description.

Table S1, the result of 441 SHW lines should not be presented as supplementary.

Line 215, please specify how many DArTSeq markers you obtained from genotyping.

Line 222, please specify the 20 MTAs in Table 2. The legend of Table should have more details.

Line 215-263, this section is not clearly presented. You put too many important information in supplementary.

Line 265-275, this is also confusing. I suggest specifying the QTL in the table with MTAs.

Line 272, please check the R2, 0.8138 seems too high.

Line 283-284, please check this sentence.

Both section 3.5 and Figure 3 need to be revised to be clearer.

Line 317-343, to me, these are introduction. If you want to keep them in discussion, they should be written with your result instead of just cite references.

Overall, the MATs and QTL you identified are minors based on the -log10 p-value and R2. However, they are what you got. Please clearly present each MATs/QTL that are significant and all the SHW you would recommend to the breeders.

Author Response

Dear sir/Madam;

Thanks for your comments. We have revised the manuscript based on your comments and asked for English language improvement. Please find response to your quesitons. We have highlighted text addressing your specific comments. 

Line 14, if two SHW lines did not germinate (line 194), there were only 441 lines that were evaluated. DONE

Line 53, you may delete “disease”. DONE

Line 56, new sources of “what”? ADDED INFORMATION

Line 71-72, what is the relevance to have this sentence? IMPORTANCE OF THIS STUDY

Line 83, still, please clear the number of lines. UPDATED and CORRECTED

Line 104, the size of containers? INFORMATION PROVIDED

Line 117-118, how many ml of suspension per seedling? How long was between each time spray? INFORMATION PROVIDED

Line 128, how many grams of leaf? Did you use the same plants for both phenotyping and genotyping? INFORMATION PROVIDED. Not the same plants but same entry.

Line 133-140, if this is the description of DArTSeq, it should be in introduction section. Please clarify the procedure you performed for genotyping. PROVIDED

Line 143-146, this should be in result section. DONE

Line 157, it is not proper to cite reference like “by [49]”. REVISED

Line 157-172, again, too much description about MLM and PLS. Please describe in detail about how you did instead of introducing them. REFERENCE PROVIDED for DETAILS.

Line 184, please replace “between” with “among”. DONE

Figure 1, I don’t think that “Frequency” and “Spot Blotch” are suitable titles. The legend should have more description. DONE

Table S1, the result of 441 SHW lines should not be presented as supplementary. TOO BIG TABLE and HENCE INCLUDED as SUPPLEMENTARY

Line 215, please specify how many DArTSeq markers you obtained from genotyping. DONE

Line 222, please specify the 20 MTAs in Table 2. The legend of Table should have more details. DONE

Line 215-263, this section is not clearly presented. You put too many important information in supplementary. REVISED

Line 265-275, this is also confusing. I suggest specifying the QTL in the table with MTAs. REVISED

Line 272, please check the R2, 0.8138 seems too high. RECHECKED and CORRECTED

Line 283-284, please check this sentence. DONE

Both section 3.5 and Figure 3 need to be revised to be clearer. DONE

Line 317-343, to me, these are introduction. If you want to keep them in discussion, they should be written with your result instead of just cite references. CHANGES MADE

Overall, the MATs and QTL you identified are minors based on the -log10 p-value and R2. However, they are what you got. Please clearly present each MATs/QTL that are significant and all the SHW you would recommend to the breeders. DONE
